# Experimental and Numerical Investigation of Axial Compression Behaviour of FRP-Confined Concrete-Core-Encased Rebar

**DOI:** 10.3390/polym15040828

**Published:** 2023-02-07

**Authors:** Jingzhou Lu, Han Huang, Yunkai Li, Tong Mou

**Affiliations:** School of Civil Engineering, Yantai University, Yantai 264005, China

**Keywords:** FRP-confined concrete-core-encased rebar, buckling, monotonic loading, axial compressive behaviour, finite element analysis

## Abstract

The axial compression behaviour of fibre-reinforced polymer (FRP)-confined concrete-core-encased rebar (FCCC-R) was investigated by performing monotonic axial compression tests on seven groups of FCCC-R specimens and three groups of pure rebar specimens. The research parameters considered were the FRP winding angle (0°, ±45°, and 90°), number of layers (2, 4, and 6 layers), and slenderness ratio of specimens (15.45, 20, and 22.73). The test results showed that FCCC-R’s axial compression behaviour improved significantly compared with pure rebar. The axial load–displacement curves of the FCCC-R specimens had a second ascending branch, and their carrying capacity and ductility were enhanced substantially. The best buckling behaviour was observed for the FRP winding angle of 90°. The capacity and ductility of the specimens were positively related to the number of FRP-wrapped layers and inversely related to the slenderness ratio of the specimens. A finite element model of FCCC-R was constructed and agreed well with the test results. The finite element model was used for parametric analysis to reveal the effect of the area ratio, FRP confinement length, internal bar eccentricity, and mortar strength on the axial compression behaviour of FCCC-R. The numerical results showed that the area ratio had the most significant impact on the axial compression behaviour of FCCC-R. The confinement length of the FRP pipe and internal bar eccentricity had similar effects on the axial compression behaviour of FCCC-R. Both of them had a significant impact on the second ascending branch, with the post-peak behaviour exhibiting minimal differences. The influence of mortar strength on the axial compression behaviour of FCCC-R was observed to be minimal.

## 1. Introduction

Located between the circum-Pacific seismic belt and the Eurasian seismic belt, China is squeezed by the Pacific plate, Indian Ocean plate, and Philippine Sea plate, resulting in frequent earthquake activities. Laboratory evidence and earthquake damage observations indicate that the buckling of longitudinal steel bars is a vital damage state of reinforced concrete (RC) columns when subjected to ever-increasing side-sway inelastic deformations [1,2,3].

For RC columns with insufficient lateral reinforcement, a large transversal reinforcement distance and large axial compression ratio and axial compressive force combined with a bending moment will produce a large strain at the section edge, and large tensile strains followed by high compression will cause the concrete to crush or spall. When the cover concrete spalls, the longitudinal reinforcement loses the support of the surrounding concrete, resulting in inelastic buckling due to insufficient restraint. The reinforcement will undergo large cyclic deformation after buckling. Early fracture of the reinforcement is likely to occur under low-cycle fatigue, resulting in the loss of the lateral bearing capacity and the collapse of the RC column [4]. Therefore, there is a need to conduct research to develop ways to increase the rebar ductility and prevent the longitudinal rebar from premature buckling. Developing such techniques can assist in the construction of structures with elevated seismic resistance performance.

To prevent premature buckling of the longitudinal rebar in reinforced concrete columns, researchers have proposed many approaches, among which the appropriate arrangement of lateral reinforcement in reinforced concrete columns has been widely studied and was accepted by concrete construction design criteria [5,6,7]. However, extra lateral reinforcement might make the steel rebar overcrowded, which is not only inconvenient for construction in the beam–column joint area but also makes it difficult to cast and vibrate concrete evenly to achieve the expected strength [4]. In addition, a hybrid reinforcing component, called a steel restraint ring, which uses a steel pipe to restrain the rebar, has also been developed to prevent the buckling of the longitudinal rebar [8,9,10]. The component works better on columns with slimmer rebar compared with regular reinforced concrete columns. With the development of fibre-reinforced polymer (FRP), it was realised that FRP had excellent properties such as high strength, fatigue resistance, and corrosion resistance. In recent years, a lot of research results have been achieved on FRP instead of steel-tube-confined concrete, and the confined specimens all showed significant compressive performance enhancement [11,12,13,14,15,16,17,18,19,20,21]. Feng et al. [22] proposed a concept in which a slender steel component was introduced into a mortar-filled FRP tube. Axial compression tests of a series of specimens showed that the specimen had better axial compressive behaviour than a pure steel component. Wang et al. [4] proposed a similar concept: FRP-confined concrete-core-encased rebar (FCCC-R). Their research showed that the constraining effect of concrete and the FRP tube effectively prevented premature buckling of the internal rebar. On the one hand, for a single FCCC-R specimen, the restraint effect of FRP improves the compressive performance of concrete or mortar and prevents the concrete cover from spalling. Under the support of the outer concrete, the buckling performance of the internal reinforcement has been greatly improved. In addition, under the double reinforcement of the FRP pipe and mortar, the section inertia moment of the internal reinforcement increases, which enhances the axial compression performance of the specimen. On the other hand, for reinforced concrete RC columns equipped with FCCC-R, the high-strength concrete or mortar filled in the FRP pipe improves the axial compressive strength of the column. In addition, the confinement of the FRP pipe can effectively improve the strength of confined concrete [23]. Under the action of seismic load, after the concrete surrounding FCCC-R in the RC column is crushed under compression, FCCC-R can still be used as a high-strength skeleton to bear the force alone, which significantly improves the anti-seismic performance of the column. In their study, a few horizontal cracks appeared on the mid-span side of the specimens. This was because the FRP winding direction adopted for all FCCC-R specimens was close to 90° to the cross-sectional direction of the specimen, which only provided better hoop constraints. Once the FRP pipe was bent, cracks easily developed on the side, resulting in the failure of the entire specimen. Therefore, it is necessary to study the effect of the FRP winding angle on the axial compression behaviour of FCCC-R.

Using the concept of FCCC-R proposed by Wang et al. [4], this study further investigated and analysed its buckling stability. Monotonic axial compression tests of seven groups of FCCC-R specimens and three control groups of pure rebar specimens were performed to analyse the axial compression behaviour and failure mode of specimens with different winding angles, winding thicknesses, and slenderness ratios. Furthermore, a finite element model was constructed to verify the test results. The finite element model was also used to perform parametric analyses to reveal the effects of the area ratio, FRP pipe confinement length, internal bar eccentricity, and mortar strength on the axial compression behaviour of FCCC-R.

## 2. Overview of Test

### 2.1. Specimen Design

The configuration of the FCCC-R specimen is shown in Figure 1. The exterior of the specimen was an FRP pipe, and the internal reinforcement was placed at the centre of the cross section of the FRP pipe. High-strength concrete or mortar was filled in the middle of the FRP pipe and internal rebar.

The length of the FRP pipe was defined as the confinement length *L*_r._ For all FCCC-R specimens, a reserve length of 90 mm at each end of the steel bar was not constrained by the FRP pipe, and out of this reserve length, *L*_a_ = 70 mm was the supporting length in the test to ensure that both ends were fixed; *L*_b_ = 20 mm was set as the length between the end of the test fixture and the end of the FRP tube to ensure that the axial load was applied only to the internal rebar. *L* = *L*_r_ + 2*L*_b_ was the loading length of the specimen. The fibre winding angle *α* was defined as the angle between the fibre and the cross section of the specimen, and it was positive in the counter-clockwise direction and negative in the clockwise direction. Furthermore, the steel rebar’s diameter *d* was 22 mm, the core concrete’s diameter *D* was 40 mm, and the thickness of the FRP pipe’s wall, *t*, was 2 mm.

### 2.2. Specimen Preparation

The preparation of the FCCC-R specimen was divided into three steps: mould making; mortar pouring, mould removal, and maintenance; and fibreglass cloth wrapping. These steps are described below.

Mould making: Made of a ready-made PVC pipe, the specimen mould was divided into the middle and the base parts, as shown in Figure 2a. To study the buckling performance of FCCC-R at different slenderness ratios, we intercepted 300, 400, and 500 mm long PVC pipes. A lubricant was then applied to these pipes to place them in the middle of the mould. A 90 mm long PVC pipe was cut to form the mould base because FCCC-R required 90 mm long rebar exposed at both ends, with the top being left outside. One end of the rebar was wrapped with soft plastic and placed in the base to ensure that the rebar was in the centre position and to prevent the mortar from being exposed at the bottom of the mould. Subsequently, the middle of the mould was connected to the base with adhesive tape, ready for mortar pouring.

Mortar pouring, mould removal, and maintenance: Mortar and water were taken in a mortar mixer and continuously mixed. The mixture was poured into the mould when the mortar fluidity met the requirement. The sidewall of the mould was tapped while pouring the mixture, and a hollow circular iron plate was used to position the upper part of the rebar. Three cubic mortar specimens were poured simultaneously. After completing this, the mould was removed, and the mortar specimens were placed in a concrete curing room for 28 days.

Fibreglass cloth wrapping: Before the mortar specimens were wrapped in fibreglass cloth, sandpaper was used to remove floating ash on their surface, and the exposed steel bars at both ends were wrapped with plastic film. Then, fibre cloth with a lap length appropriate for the test study parameters was cut. A binder was evenly applied to the outer surface of the mortar and rested for a period of time. After the outer surface of the mortar had dried, the fibre cloth was pasted on it. Finally, the specimens were placed in a dry environment for seven days for curing and reinforcing. The reinforced FCCC-R specimens are shown in Figure 2b.

### 2.3. Test Materials

#### 2.3.1. Rebar

In this test, HRB400 rebar with a diameter of 22 mm was used. Tensile tests were performed on it according to GB/T228.1-2010, Tensile Test of Metal Materials at Room Temperature Part 1: Greenhouse Test Method. The measured material property indexes of the rebar are presented in Table 1.

#### 2.3.2. Mortar

In this test, in order to better cast and form FCCC-R, mortar with the same strength as concrete but with better fluidity was selected. The mortar material was high-strength non-shrink grouting material produced by Qingdao ZhuoNengda Construction Technology Co., Ltd. (Qingdao, China) Samples required for the material performance test were prepared according to GB/T50081-2019, Standard for test methods for physical and mechanical properties of concrete, and the method presented in the literature [4]. After preparation, the specimens were placed in a curing room at a constant temperature and humidity. After 24 h, the mould was shifted to a standard concrete curing room, which was kept for 28 days. The WE-1000 hydraulic universal testing machine was used to perform the loading test on the mortar specimens, and the cubic compressive strength of the mortar specimens was measured to be 43.59 MPa. In order to better perform the finite element analysis of the specimen, the data were converted into cylinder axial compressive strength. The material property indexes of the mortar are presented in Table 2.

#### 2.3.3. FRP

FRP was made of unidirectional glass fibre cloth and epoxy resin glue produced by Hai Ning Anjie Co., Ltd. After the fibre sheet was prepared, the tensile test was performed according to GB/T3354-2014, Test method for tensile properties of oriented fibre-reinforced polymer matrix composites. Furthermore, the longitudinal and horizontal shear test was performed according to GB/T3856-2005, Longitudinal and transverse shear test method for polymer matrix composites. The mechanical property indexes of composite materials are presented in Table 3.

### 2.4. Test Method

In order to study the effects of the FRP winding angle, winding thickness, and slenderness ratio of the specimen on the buckling performance of FCCC-R under an axial compressive load, the following steps were followed. The number of FRP constraint layers was divided into internal and external layers, with the number of both layers being equal. All FRP external layers were circumferentially wrapped, that is, *α* = 0°, to protect the specimen from developing longitudinal cracks because of the insufficient hoop constraint. In this part, the internal layer winding angles were varied to study the effect of the FRP winding angle, the number of FRP layers was varied to study the effect of the winding thickness, and the length of the FRP pipe was varied to study the effect of the slenderness ratio of the specimen.

Monotonic axial compression tests were performed on seven groups of FCCC-R specimens and three groups of pure rebar specimens, with each group of specimens containing three identical specimens to avoid behavioural deviation caused by the rebars’ initial defect and initial load eccentricity. The monotonic loading specimens are listed in Table 4. The nomenclature of the specimen groups is as follows: for FCCC-R specimens, the group name begins with “F”, followed by a number indicating the length of the FRP restraint. The number after the slash indicates the loading length of the specimen. The numbers ‘2′, ‘4′, and ‘6′ indicate the number of winding layers of FRP, and the letters ‘a’, ‘b’, and ‘c’ indicate fibre winding angles of ‘0°’, ‘±45°’, and ‘90°’ for the internal layer, respectively. For normal rebar, the group name starts with “R” and the number after it indicates the loading length of the rebar.

All the specimens were pressurised on an SDS500 electro-hydraulic servo dynamic and static universal testing machine, and the maximum range of the testing machine was 500 kN. In the test, after switching to the displacement loading mode, we set the loading amplitude to 1 mm/min and recorded the load–displacement relationship of the specimens.

The test setup of the FCCC-R specimen is illustrated in Figure 3. During the axial compression test, the test setup was calibrated vertically to ensure that the specimen could be damaged under axial compression and reduce the eccentric pressure damage as much as possible. The two fixtures of the testing machine provided fixed-end conditions during the test. The mean value of the result obtained by the two vertical LVDTs was used to eliminate the influence of the slight rotation of the clamps that occurred during the test.

## 3. Test Result and Analysis

### 3.1. Test Phenomena and Mechanism Analysis

#### 3.1.1. Rebar Buckling

In the initial stage, the load increased monotonically with the loading time. Soon, it reached the peak, and it subsequently decreased rapidly. At this time, it was observed that the steel bar was rapidly bent, resulting in the “buckling” phenomenon, and the test was terminated to ensure safety. The specimens R-340, R440, and R540 all buckled before reaching the yield load. The reason is that rebar buckles before reaching the yield load when the slenderness ratio exceeds 9 or 10, and the slenderness ratios of all three groups of specimens exceeded 10.

#### 3.1.2. Mechanism Analysis of FCCC-R Buckling Failure

In the beginning, the FCCC-R specimens remained elastic, and the relationship between the axial load and displacement was basically linear. As the force acting on the FRP cloth and the axial strain value was relatively small, the restraint effect was not obvious. As the axial load increased, the strain of the FRP cloth started increasing significantly, with the strain rate gradually accelerating. The acceleration was because the appearance and development of concrete micro-cracks at this time increased the lateral deflection of the specimen, resulting in an increase in the horizontal restraint of the FRP cloth. When the stiffness of the specimen decreased to a smaller value, the load–displacement curve exhibited a second rising stage where it reached a peak. At this stage, the FRP-confined area became slightly curved. Shortly after the peak load, the FCCC-R specimen buckled, the lateral displacement increased significantly, and transverse cracks appeared rapidly with an obvious cracking sound. The FRP outer layer fractured suddenly, and the bearing capacity of the specimen plummeted [24]. Simultaneously, the axial load decreased rapidly, entering the softening stage. Finally, the specimen was destroyed due to the loss of carrying capacity and excessive lateral displacement.

The overall buckling failure mode of the FCCC-R specimens is shown in Figure 4. The global buckling failure occurred in all specimens, and circumferential cracks appeared in the middle of the specimens. During testing, it was found that the fibre winding angle affected the number and morphology of cracks. Multiple circumferential cracks appeared in the middle of the specimens with an inner layer winding angle of 0°, while the middle of the specimens with inner layer fibre winding angles of ±45°and 90° mainly developed a circumferential crack, with a few smaller circumferential cracks appearing on both sides of the main crack. The explanation for this is as follows. When the outer and inner fibres of the FCCC-R specimens were all circumferentially wound, the entire FRP pipe had a strong circumferential restraint, and the longitudinal tension came only from the adhesive force associated with the fibre glue. Once the specimen buckled, multiple circumferential cracks were rapidly formed. When the inner fibre winding angle was ±45°or 90°, the entire FRP pipe not only produced a certain hoop restraint but also provided a certain longitudinal tension, which reduced the formation of hoop cracks.

On some specimens, diagonal cracks appeared at the edge of the mortar. Local failure diagrams of the FCCC-R specimens are shown in Figure 5. During the loading process of the specimens, the fibres at the ends of one F-400/440-2-c specimen and one F-300/340-4-c specimen were torn longitudinally, and the load dropped rapidly before reaching the ideal value. The reason was that the rebar was incorrectly positioned. It was not placed in the middle of the specimen during specimen production, resulting in insufficient local hoop restraint which caused the formation of longitudinal cracks.

### 3.2. Test Result Comparison and Analysis

#### 3.2.1. Summary of Test Results

The buckling test results of the specimens are presented in Table 5. *P*_max_ is the buckling peak load of the specimen, *P*_max,s_ is the buckling peak load of the rebar with the same slenderness ratio, *U*_max_ is the axial displacement corresponding to *P*_max_, and *F*_y_ is the yield load of the steel rebar. The values of *P*_max_*/P*_max,s_ are greater than 1, and it can be seen that the FCCC-R specimens improve the compressive performance of steel bars to different extents. The values of *P*_max_*/F*_y_ showed that the pure steel rebars suffered buckling failure before reaching the yield strength, while the steel bars of the FCCC-R specimens yielded when buckling instability occurred.

#### 3.2.2. Analysis of Effect of Winding Angle on FCCC-R Axial Compression Performance

It is evident from Table 5 and Figure 6a that when the thickness of the fibre layer and the slenderness ratio were constant, all FCCC-R specimens, irrespective of the fibre winding angle, had a second ascending branch, and their carrying capacity and ductility appeared to be better than those of a pure steel bar with the same slenderness ratio. The peak loads of the FCCC-R specimens with inner fibre winding angles of 0°, ±45°, and 90° were 31%, 39%, and 48% higher than those of the corresponding pure steel bars, respectively. The axial displacements corresponding to the peak loads were 218%, 323%, and 390% higher than those of the corresponding pure steel bars, respectively.

Therefore, when the thickness of the fibre layer and the slenderness ratio were constant, the optimal winding angle of the inner layer fibre was 90°. This was because when the inner fibre was wound at 90°, the entire FRP tube not only exerted a certain hoop restraint on the internal rebar but also provided a certain longitudinal tension, thereby reducing the generation of circumferential cracks and slowing down the occurrence of buckling. However, if it is hoop winding, the FRP hoop restraint is very strong. Once buckling occurs, multiple hoop cracks appear in the specimen, which accelerates the occurrence of buckling.

#### 3.2.3. Analysis of Effect of Winding Thickness on FCCC-R Axial Compression Performance

From Table 5 and Figure 6b, it is apparent that when the fibre winding angle and slenderness ratio were constant, there was generally no difference in the first ascending branch between the FCCC-R specimens with different fibre winding thicknesses, but the length of the second ascending branch was different. The longer the fibre winding thickness, the longer the length of the second ascending branch. The data of Table 5 show that the peak loads of the FCCC-R specimens with 2, 4, and 6 fibre winding layers were 30%, 48%, and 68% higher than those of the corresponding pure steel bars, respectively. The axial displacements corresponding to the peak loads were 214%, 390%, and 576% higher than those of the corresponding pure steel bars, respectively.

Therefore, when the fibre winding angle and slenderness ratio were constant, the peak load and ductility of the FCCC-R specimens increased gradually with an increase in the FRP winding thickness. This was because with an increase in the FRP winding thickness, the restraint and longitudinal tension provided by the FRP increased, which limited the transverse deformation of the specimens [25] and thereby slowed down the occurrence of buckling.

#### 3.2.4. Analysis of Effect of Slenderness Ratio on FCCC-R Axial Compression Performance

It is evident from Table 5 and Figure 6c that when the fibre winding angle and winding thickness were constant, FCCC-R specimens with different slenderness ratios differed in terms of the second ascending branch. The larger the slenderness ratio, the shorter the length of the second ascending branch, while the linear stage almost remained unchanged. The data of Table 5 show that the peak loads of the FCCC-R specimens with slenderness ratios of 22.73, 20, and 15.45 were 15%, 31%, and 53% higher than those of the corresponding pure steel bars, respectively. The axial displacements corresponding to the peak loads were 188%, 390%, and 760% higher than those of the corresponding pure steel bars, respectively. In addition, the slenderness ratio had a greater impact on the FCCC-R specimens than on the pure steel bars, as evident from Table 5, and it had a more significant effect on the FCCC-R axial compression performance than the FRP winding angle and thickness.

Therefore, when the fibre winding angle and winding thickness were constant, the peak load and ductility of the FCCC-R specimens decreased as the slenderness ratio increased. There were two reasons for this: first, with an increase in the slenderness ratio, the non-uniformity of the circumferential strain distribution of the FRP cloth in the height direction increased, and the lateral restraint effect decreased, which could not ensure the high performance of the axial compression of the specimens and hence led to a decrease in the bearing capacity of the specimens. Second, the lateral deflection of the specimens increased with the slenderness ratio. It is noted that for specimens with a slenderness ratio greater than 12.5, lateral deflection occurs at the initial loading stage, and it gradually changes from linear growth to nonlinear growth with an increase in the load. When the load is on the verge of reaching the ultimate load, the lateral deflection of the specimens increases sharply. Finally, instability and failure occur because of excessive lateral deflection [26]. Therefore, the greater the slenderness ratio of the specimen, the greater the lateral deflection, which accelerates the occurrence of buckling.

## 4. Finite Element Simulation and Parametric Analysis

### 4.1. Material Constitutive Model

For the steel rebars, the elastic–plastic trilinear constitutive model with a yield plateau was adopted, and for the mortar materials, we used the concrete damage plastic model in [4] and its compression and tension constitutive relations. As for the FRP materials, the mechanical behaviour of FRP was divided into elastic and damage stages, and lamina elastic material was used in the elastic stage [27,28,29]. The Hashin damage criterion was used to define the fracture failure of the composite materials in the damage stage [30,31,32]. Specific data are shown in Table 3.

### 4.2. Establishment of Finite Element Model

Since the FCCC-R specimens comprised an internal steel bar, mortar, and an FRP pipe, the solid element and shell element were used in ABAQUS to separately establish these three components, as shown in Figure 7a–c. Among them, the C3D8R solid element was used for the mortar and internal steel bar, and the S4R shell element was used for the FRP pipe because its thickness was very small compared with the cross section of FCCC-R. The three components were assembled and grids were divided to integrate the entire FCCC-R specimen, as shown in Figure 7d.

### 4.3. Verification Result of Finite Element

#### 4.3.1. Failure Mode

As shown in the experimental results, there were two main failure modes of FCCC-R. First, owing to the circumferential winding of the inner layer fibre, the tension in the longitudinal direction was provided only by the fibre glue. However, the tensile strength of a colloid is low. Therefore, there were multiple circumferential cracks in the middle of the specimens, causing damage to the specimens. Second, the angled winding of the inner fibre led to the fibre providing a certain longitudinal tension, and the outer composite colloid was subjected to the longitudinal tensile action of the inner fibre. Therefore, the failure of the specimen was circumferential crack propagation. ABAQUS simulated the two failure modes well, and its simulations were consistent with the test results. The simulation and experimental failure modes of the F-400/440-4-a and F-400/440-4-c specimens are shown in Figure 8.

#### 4.3.2. Load–Displacement Curve

ABAQUS was used to simulate the seven groups of FCCC-R specimens under monotonic axial compression, and the load–displacement curves obtained are shown in Figure 9. Clearly, the match between simulation and experimental load–displacement curves is very good at the elastic stage, and the curves show the same trend after entering the elastic–plastic stage.

The test and simulated values of the peak load for each specimen are listed in Table 6. *P*_max_ is the peak load measured in the test, and *P*_max,M_ is the peak load calculated by the simulation. It can be seen from the results that the maximum error between the two values is 7.30%, and the average ratio is 1.03. The simulation results are in good agreement with the test results, indicating that the finite element model can simulate the load–displacement relationship of the FCCC-R specimens under monotonic axial compression accurately.

### 4.4. Parametric Study and Analysis

In order to further study the factors affecting the axial compression performance of FCCC-R and make up for the limitations of the objective conditions in the test, on the basis of the finite element model constructed, the influence parameters that could not be studied in the test were analysed. The basic dimensions of the finite element model were determined based on the F-400/440-4-a group. The parameters studied in this section were the area ratio *A/A_s_*, confinement length *L_r_/L,* internal bar position eccentricity *e/R*, and mortar strength *f_c_.* Table 7 lists the ranges of all parameters considered in the parametric study.

The area ratio *A/A_s_* of FCCC-R is defined as the ratio of the FRP pipe’s internal area *A* to the internal bar’s cross-sectional area *A_s_*. As shown in Figure 10a, the peak stress and average strain increased dramatically with an increase in the area ratio. For a small area ratio, such as the area ratio of 2.53, there was no second ascending branch, and the axial load decreased immediately after the peak load. This was because the smaller the area ratio, the smaller the moment of inertia of the section provided, which was insufficient to resist the influence of the additional moment caused by the lateral displacement. The additional moment in the middle of the span of the specimen was too large, and hence the rebar buckled, causing the premature failure of the specimen. Thus, a large area ratio is preferable in the design of the FCCC-R specimen.

FCCC-R’s confinement length ratio *L_r_/L* is defined as the ratio of the FRP pipe’s length *L_r_* to the internal bar’s length *L*. If FCCC-R is placed in a concrete column, the steel bar may not be entirely confined by the FRP pipe because of the consideration of the steel bar anchorage and connection. The two ends of the steel bar are intended to remain outside the FRP pipe. Hence, it is necessary to investigate the influence of confinement length. As shown in Figure 10b, the specimens’ bearing capacity and ductility improved to a certain extent with an increase in the confinement length, and the greater the confinement length ratio, the more obvious the improvement was. This was because a longer FRP confinement length resulted in a stronger constraint action and higher longitudinal tension on the specimen, which resulted in poorer circumferential deformation and slowed down the occurrence of buckling. Thus, a large restrained length ratio is preferable in the design of FCCC-R specimens.

The internal bar position eccentricity *e/R* of FCCC-R is defined as the ratio of the distance *e* from the centre of the inner reinforcement to the centre of the FRP pipe to the inner radius *R* of the FRP pipe. It can be found from Figure 10c that the specimens’ axial compressive behaviour gradually deteriorated and their bearing capacity along with ductility continued to decrease as the bar’s eccentricity increased. This can be explained as follows: with an increase in the internal bar eccentricity, the restraint effect of FRP and concrete on one side of the reinforcement is smaller than that at the central position, which accelerates the occurrence of buckling. Therefore, a better configuration of a single FCCC-R can be obtained by placing the rebar at the centre of the cross section under axial compressive loads.

Figure 10d shows the effect of the mortar strength on FCCC-R’s axial compression performance. The peak load slightly increases as the mortar strength increases, and its influence is much smaller compared with that of other parameters. However, the behaviour after the peak is almost the same. The reason was that with the mortar cracking under tension, all specimens tended to fail without the mortar being crushed. Thus, the mortar provided only tensile strength, and the tensile strength of the mortar was very low. Therefore, the use of mortars with different strengths had little impact on the axial compression performance of FCCC-R.

## 5. Conclusions

In this study, the buckling performance of FCCC-R under monotonic axial compression was analysed by performing the monotonic buckling test and a simulation on commercial finite element software. The main conclusions are as follows:

(i) The axial compression performance of FCCC-R was found to be superior to that of pure steel bars due to the dual constraints of FRP and concrete. The load–displacement curve of FCCC-R demonstrated a second ascending branch, characterised by high bearing capacity and ductility. As a result, the internal steel bars with a high slenderness ratio were able to reach or surpass the yield strength during buckling, effectively avoiding premature buckling under compression.

(ii) The experimental results indicated that the most optimal fibre winding angle for FCCC-R was 90°. The axial compression performance of FCCC-R was observed to improve with an increase in the thickness of FRP winding. Moreover, the slenderness ratio was identified as the primary factor affecting the axial compression performance of FCCC-R. A reduction in the slenderness ratio resulted in a noticeable increase in the length of the second ascending branch in the load–displacement curve. To be specific, a decrease in the slenderness ratio from 22.73 to 15.45 resulted in an increase in the bearing capacity and ductility of the specimens by 33% and 175%, respectively.

(iii) A finite element model of FCCC-R was established using the ABAQUS software. The model was validated through comparison with experimental results, with a maximum discrepancy of 7.30%. The simulation results were found to be in accordance with the experimental outcomes, indicating the reliability and validity of the established model.

(iv) A parametric analysis was conducted, revealing that the area ratio is the most critical factor that affects the axial compression performance of FCCC-R. As such, it is advisable to increase the area ratio in the design of FCCC-R. Moreover, improvements in the axial compression behaviour of FCCC-R can be achieved by increasing the FRP confinement length and reducing the eccentricity of reinforcement. In particular, it was observed that the mortar strength had a minimal impact on the axial compression performance of FCCC-R.

## Figures and Tables

**Figure 1 polymers-15-00828-f001:**
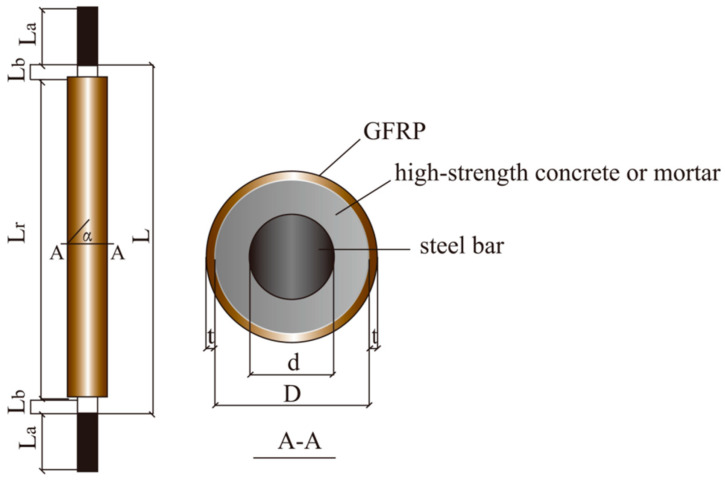
Configuration of fibre-reinforced-polymer-confined concrete-core-encased rebar (FCCC-R) specimens.

**Figure 2 polymers-15-00828-f002:**
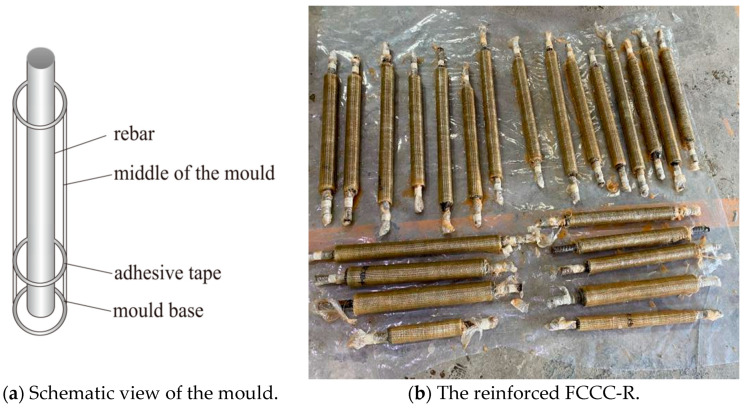
Schematic view of the mould and reinforced FCCC-R specimens.

**Figure 3 polymers-15-00828-f003:**
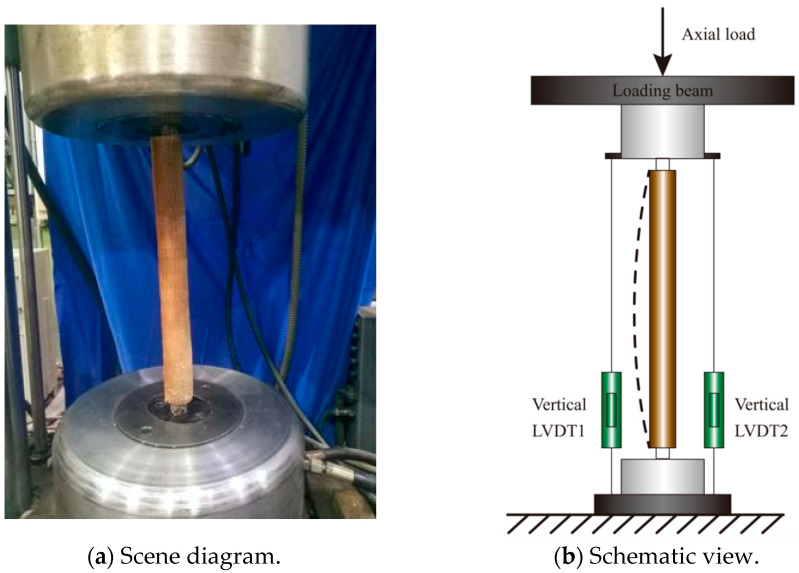
Test setup of FCCC-R specimen.

**Figure 4 polymers-15-00828-f004:**
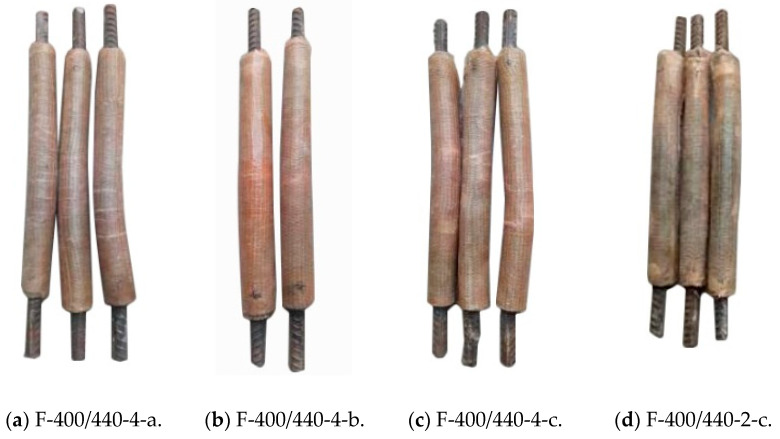
Global buckling failure mode of FCCC-R specimens.

**Figure 5 polymers-15-00828-f005:**
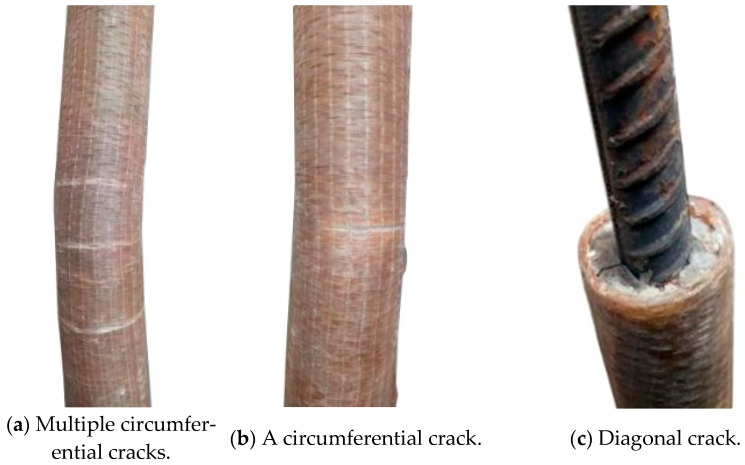
Local failure diagrams for FCCC-R specimens.

**Figure 6 polymers-15-00828-f006:**
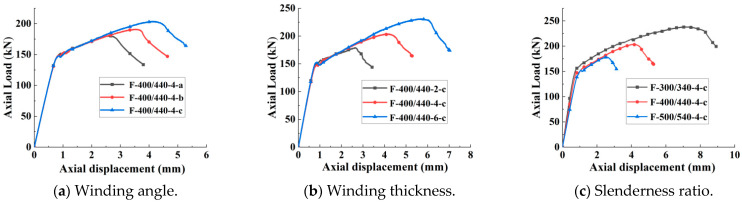
Comparison of load–displacement relationships between different groups.

**Figure 7 polymers-15-00828-f007:**
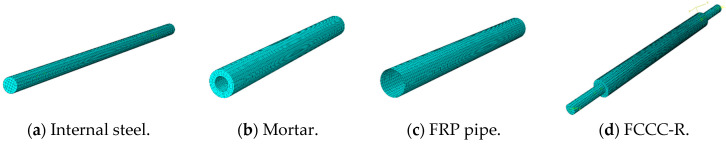
Configuration of FCCC-R model.

**Figure 8 polymers-15-00828-f008:**
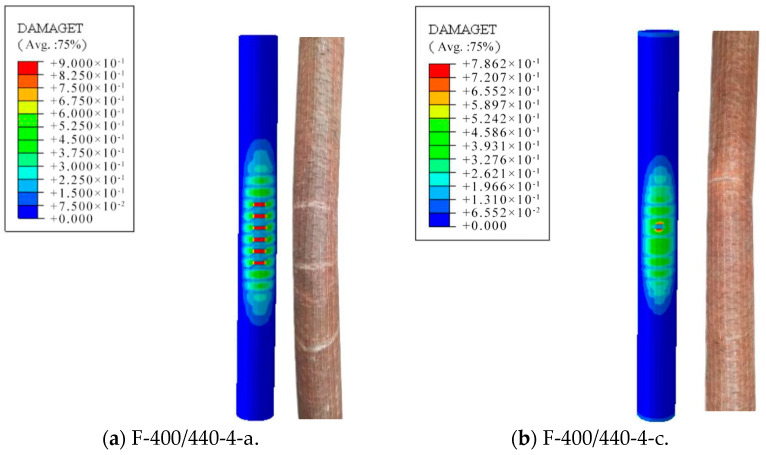
Comparison between simulated and experimental failure modes of FCCC-R specimens.

**Figure 9 polymers-15-00828-f009:**
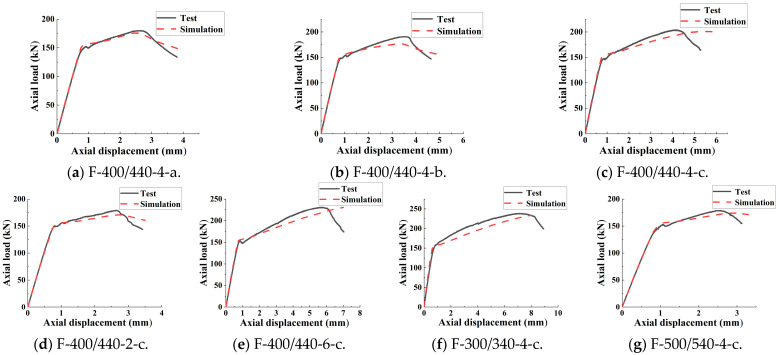
Load–displacement curve comparison for FCCC-R specimens.

**Figure 10 polymers-15-00828-f010:**
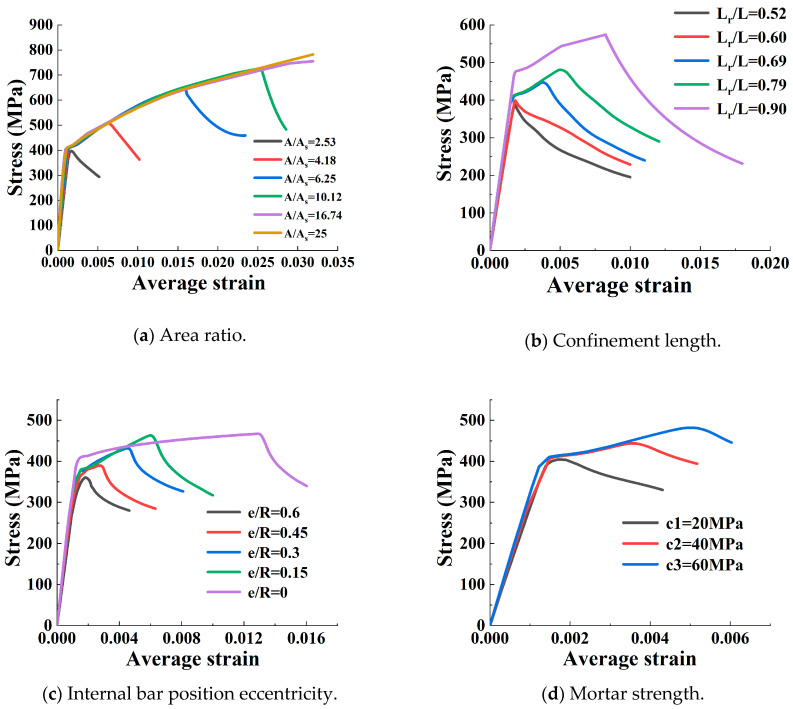
Effect of different parameters on the average stress–strain relationships under compression.

**Table 1 polymers-15-00828-t001:** Material property indexes of rebar.

Rebar Type	Yield Strength (MPa)	Yield Strain	Tensile Strength (MPa)	Elastic Modulus (GPa)	Poisson’s Ratio
HRB400	409	0.00207	566	196.7	0.3

**Table 2 polymers-15-00828-t002:** Material property indexes of mortar.

Coupon Shape	Axial Compressive Strength (MPa)	Elastic Modulus (GPa)	Poisson’s Ratio
Cylinder	34.44	27.76	0.21

**Table 3 polymers-15-00828-t003:** Mechanical property indexes of composite materials.

Longitudinal Tensile Strength (MPa)	Transverse Tensile Strength (MPa)	Shear Strength (MPa)	Longitudinal Elastic Modulus (GPa)	Transverse Elastic Modulus (GPa)	Shear Modulus (GPa)	Poisson’s Ratio
760.2	55.1	217.1	45.1	2.7	14.9	0.23

**Table 4 polymers-15-00828-t004:** Monotonic loading specimens.

Specimen ID	Confinement Length (mm)	Loading Length (mm)	Layers	Confinement Thickness (mm)	Angle (from Inside to Outside)
F-400/440-4-a	400	440	4	2.8	0°0°0°0°
F-400/440-4-b	400	440	4	2.8	−45° + 45°0°0°
F-400/440-4-c	400	440	4	2.8	90°90°0°0°
F-400/440-2-c	400	440	2	1.4	90°0°
F-400/440-6-c	400	440	6	2.4	90°90°90°0°0°0°
F-300/340-4-c	300	340	4	2.8	90°90°0°0°
F-500/540-4-c	500	540	4	2.8	90°90°0°0°
R-340	--	340	--	--	--
R-440	--	440	--	--	--
R-540	--	540	--	--	--

**Table 5 polymers-15-00828-t005:** Test results for specimens.

Specimen ID	*L/d*	*P*_max_ (kN)	*U*_max_ (mm)	*P* _max_ */F* _y_	*P* _max_ */P* _max,s_
F-400/440-4-a	20	180.16	2.67	1.16	1.31
F-400/440-4-b	20	190.90	3.55	1.23	1.39
F-400/440-4-c	20	203.69	4.12	1.31	1.48
F-400/440-2-c	20	178.83	2.64	1.15	1.30
F-400/440-6-c	20	231.00	5.68	1.49	1.68
F-300/340-4-c	15.45	238.17	7.05	1.53	1.62
F-500/540-4-c	22.73	178.55	2.56	1.15	1.39
R-340	15.45	146.63	0.82	0.94	
R-440	20	137.63	0.84	0.89	
R-540	22.73	128.42	0.89	0.83	

**Table 6 polymers-15-00828-t006:** Comparison between the test value and the simulation value of peak loads.

Specimen ID	*P_max_* (kN)	*P_max,M_* (kN)	*P_max_/P_max,M_*	Discrepancy
F-400/440-4-a	180.16	176.00	1.02	2.31%
F-400/440-4-b	190.90	176.97	1.08	7.30%
F-400/440-4-c	203.69	200.84	1.01	1.40%
F-400/440-2-c	178.83	170.64	1.05	4.58%
F-400/440-6-c	231.00	230.78	1.00	0.10%
F-300/340-4-c	238.17	233.48	1.02	1.97%
F-500/540-4-c	178.55	173.98	1.03	2.56%

**Table 7 polymers-15-00828-t007:** Values of the parameters considered in the parametric study.

Parameter	Values
Area ratio, *A*/*A_s_*	2.53, 4.18, 6.25, 10.12, 16.74, 25
Confinement length, *L_r_*/*L*	0.52, 0.60, 0.69, 0.79, 0.90
Internal bar position eccentricity, *e*/*R*	0.6, 0.45, 0.30, 0.15, 0
Mortar strength, *f_c_* (MPa)	20, 40, 60

## Data Availability

Data sharing is not applicable to this article.

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
