# Peer review of "Experimental and Numerical Investigation of Axial Compression Behaviour of FRP-Confined Concrete-Core-Encased Rebar"

_polymers, 2023, doi:10.3390/polym15040828_

Round 1

Author Response

Please see the attchment.

Reviewer 2 Report

First of all, the manuscript should be prepared and improved more carefully. For example, the format of references does not fit the journal's requirements.

The introduction should be revised to focus more on scientific problems.

Please add one more figure to show the test setup.

To discuss the mechanisms, one is FRP confined mortar (please refer to reference "Strength enhancement due to FRP confinement for coarse aggregate-free concretes"), and the other one is FRP confined steel reinforcement (please refer to reference "New types of steel-FRP composite bar with round steel bar inner core: Mechanical properties and bonding performances in concrete").

Please fit the simulated image and test image with the same scales in Fig. 7.

Round 2

Reviewer 1 Report

Looks good for publishment.

Author Response

Dear Reviewer,

Thank you very much for reviewing and approving our revised manuscript. We have tried our best to make minor revisions. We hope that the revised manuscript can be accepted by <Polymers>.

Thank you and best regards.

Sincerely yours,

Lujingzhou and HuangHan